# Self-Location Based on Grid-like Representations for Artificial Agents

Chuanjin Dai and Lijin Xie *

Information and Navigation School, Air Force Engineering University, Xi'an 710077, China; dcjdai@163.com
* Correspondence: major_uestc@alu.uestc.edu.cn

**Abstract:** Self-location plays a crucial role in a framework of autonomous navigation, especially in a GNSS/radio-denied environment. At the current time, self-location for artificial agents still has to resort to the visual and laser technologies in the framework of deep neural networks, which cannot model the environments effectively, especially in some dynamic and complex scenes. Instead, researchers have attempted to transplant the navigation principle of mammals into artificial intelligence (AI) fields. As a kind of mammalian neuron, the grid cells are believed to provide a context-independent spatial metric and update the representation of self-location. By exploiting the mechanism of grid cells, we adopt the oscillatory interference model for location encoding. Furthermore, in the process of location decoding, the capacity of autonomous navigation is extended to a significantly wide range without the phase ambiguity, based on a multi-scale periodic representation mechanism supported by a step-wise phase unwrapping algorithm. Compared with the previous methods, the proposed grid-like self-location can achieve a much wider spatial range without the limitation imposed by the spatial scales of grid cells. It is also able to suppress the phase noise efficiently. The proposed method is validated by simulation results.

**Keywords:** autonomous navigation; grid cell; vector navigation; velocity-controlled oscillator (VCO); phase ambiguity



## 1. Introduction

With the development of artificial intelligence (AI), artificial agents (AAs) are expected to fulfill a large number of challenging tasks instead of human beings, especially in dangerous or tough environments. As a premise, AAs should be qualified with the ability of autonomous navigation [1]. Thus, self-location is widely studied as a fundamental requirement. Current solutions usually resort to the global navigation satellite system (GNSS) and inertial navigation system (INS) [2]. Additionally, in a GNSS/radio-denied environment, vision/laser-aided localization has become the dominant method [3,4]. However, such approaches always lead to an extraordinary computational burden and fail to achieve an effective environment mapping. One issue is that since the conventional vision/laser-aided techniques deeply depend on a deep reinforcement learning (DRL) framework, they may require a large number of interactions (i.e., high-dimension input images), especially when the surrounding environment changes dynamically. Thus, data inefficiency may be inevitable, which leads to poor convergence and a heavy time cost of the training process of DRL. In addition, the DRL navigation algorithms often suffer from poor generalization and are difficult to adapt to real environments that are significantly different from the training environment.

Fortunately, the breakthroughs of brain science provide a new perspective to think about this issue. Studies have discovered that mammals can autonomously navigate depending on an internal neural representation of space, known as a "cognitive map" [5]. Further findings suggest that autonomous navigation of mammals is mainly determined by several kinds of neurons: place cells represent unique locations by a topological strategy [6]; grid cells provide a spatial metric by encoding animal's location in the two-dimensional

plane [7]; head-direction cells' activities are related to the current head orientation [8]; and boundary cells signal the boundary presence at a greater distance [9]. Since the capacity of place cells is limited for large-scale vector navigation without reinforcement learning, the cell grids have drawn the most attention, owing to the ability of "context-independent path integration" [10,11]. By taking advantage of periodic firing patterns, grid cells can update the representation of self-location by a vector, describing the animal's recent motion, which implies that the autonomous navigation can be achieved without any external signal source, such as GNSS.

Grid cells are organized into functional modules within the medial entorhinal cortex (mEC), a key part of the brain. Firing patterns of proximate grid cells demonstrate the same spatial scales but a fixed spatial offset. Moreover, grid scale increases between modules in discontinuous steps along the dorso-ventral axis of the mEC [12], which guarantees a believable localization without phase ambiguity. In engineering applications, it is assumed that the spatial firing pattern of grid cells is driven by an oscillatory interference model comprised of multiple velocity-controlled oscillators (VCOs) with different preferred directions [13]. The temporal phase of each VCO that results from path integration implies the mapped displacement along its preferred direction. Obviously, a moving mammal can be positioned by any two or more VCOs with different preferred directions, thereby actually building a framework of vector navigation. Accordingly, patterns of the oscillatory interference model are summarized and validated in [14]. The phase noise controlling mechanism is also revealed in [15]. However, the oscillatory interference model itself cannot address the large-scale navigation problem because of phase ambiguity. By resorting to Fourier shift theorem, researchers modeled the large-scale navigation through several potential neural network implementations [16]. These methods often result in extremely high computational consumption, which is not affordable in conventional AAs. Ref. [17] proposes the modular arithmetic scheme, and found that the grid-like autonomous navigation can achieve a remarkably wide spatial range. However, the spatial range is restricted within the product of module scales and the involved scales must be coprime integers. In addition, the existing research mainly focused on the biological mechanism rather than the engineering applications.

As part of an AI framework, we consider the implementation of grid-like modules in the perspective of algorithms, which will be a bridge between the biological motivation and electronic realization. In the proposed method, the performance of autonomous navigation only depends on the selection of spatial scales of grid-cells and the phase noise. The long-scale navigation can be really achieved without any inherent limitation.

The rest of this paper is organized as follows. Firstly, the idea of grid-like self-location is considered for incorporation into conventional AI architecture in Section 2, which may enhance the AA's performance of advanced behaviors. As detailed in Section 3, the spatial representation system is built by introducing a velocity-controlled oscillator (VCO) model, which performs the encoding process of self-location. To model the decoding function of grid cells, in Section 4, instead of Fourier shift theorem [16], a novel step-wise phase unwrapping algorithm is adopted to relieve the phase ambiguity in an electronic manner, with much lower computational cost. In Section 5, simulation results are provided and analyzed, followed by a concluding summary in Section 6.

## 2. Grid-Like Vector Navigation in AI Networks

Due to the significant advantages of path integration, we attempt to leverage the computational functions of grid cells to improve the navigation ability of artificial agents. It is demonstrated that grid cells can implement self-location by offering the mammals a Euclidean spatial metric and associated vector operations, known as vector navigation [18]. As previously mentioned, within each grid module in the mEC, firing patterns of grid cells share the same scale and orientation but a fixed spatial offset relative to one another; that is, an arbitrary location can be encoded by any pair of grid cells from the same module.

As argued before, owing to the periodicity of firing patterns, the mapping from real locations to grid cell representations can be sketched as the spatial phases. However, this

may lead to position ambiguity. To address this issue, several grid modules with different scales are engaged in this biological mechanism. Assume $M$ modules are adequate, with discontinuously increased scales $G_i$, i.e., $G_1$ is the smallest and $G_M$ is the largest. Within a certain module, any location can be encoded as $\varphi_1$, $\varphi_2$ with a pair of non-collinear axes 1 and 2 resulting from the previously mentioned fixed spatial offset. Thus, any location can be encoded as two sets of phases $\varphi_1 = \left\{ \varphi_{\{1,1\}}, \varphi_{\{1,2\}}, \cdots, \varphi_{\{1,M\}} \right\}$, and $\varphi_2 = \left\{ \varphi_{\{2,1\}}, \varphi_{\{2,2\}}, \cdots, \varphi_{\{2,M\}} \right\}$. It is worth noting that the angle between axis 1 and axis 2 is not necessarily $\pi/2$. Assuming a pair of encoded locations $a$ and $b$, the corresponding displacement vector can be represented as the difference between $\{\varphi_1(a)\}$, $\{\varphi_2(a)\}$, and $\{\varphi_1(b)\}$, $\{\varphi_2(b)\}$. Then, the problem of vector navigation is to recover the displacement vector. In other words, the calculation of displacement can be treated as the process of decoding a multi-scale periodic code without ambiguity. Details about the encoding and decoding mechanisms for self-location are considered in the following sections.

Motivated by the biological foundation, we consider transplanting the computational function of grid cells into the AI framework. For an artificial agent, navigation is regarded as a critical technology to adapt to an environment and the premise of other advanced behaviours. In our opinion, a grid-like module can be implemented by electronic circuits and the autonomous navigation can be modeled as a Markov decision process (MDP). To be specific, at the fixed time step, once the state information of the environment is received, the self-location can be implemented according to the previously known location with high efficiency by one-shot learning. Then, the information of current displacement is provided to the policy architecture of action control. Action will be taken to maximize the action-value function based on the current state and cumulative rewards. For the sake of clarity, we take an actor-critic (AC) [19] centered architecture as an example. The AC learner is composed of a critic and an actor. The actor, the policy network, is responsible for action selection. Meanwhile, the critic, the value network, is used to evaluate the advantage of the action taken by the actor. As shown in Figure 1, the information of velocity and heading direction is collected by corresponding sensors and passed into the grid-like module for further processing; the outputs of grid-like module are provided as the current state information for the AC learner, forming the network rewards. The vision module performs in a similar way, but concentrates on the obstacle avoidance and goal indication. In other words, the vector navigation performed by the grid-like module is adopted as the main approach to simultaneous localization and mapping (SLAM). It is worth noting that the intrinsic reward is deduced by the critic within the AC learner, and the extrinsic reward is deduced by external sources, such as a vision module, which is also provided to the actor.

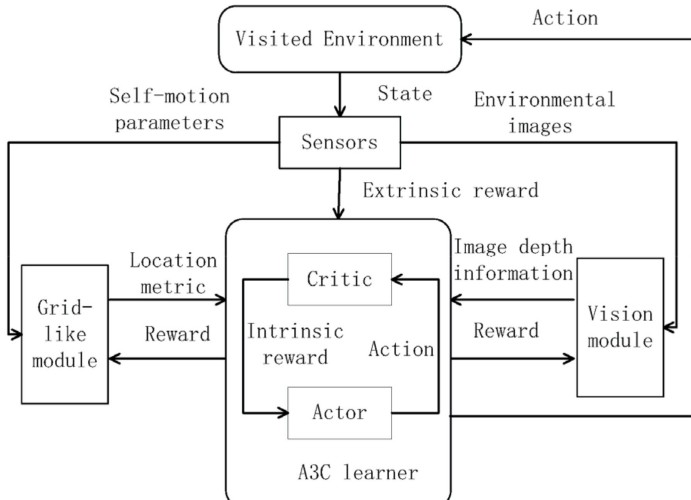

**Figure 1.** AC-centered architecture with a grid-like module.

### 3. Encoding Scheme for Grid-Like Self-Location

Because of the periodicity, the firing pattern of grid cells can be quantified as phase values, i.e., the location encoding. For a typical artificial agent, location encoding is actually the dynamic response to environment state change. Hence, a computational model of grid cell firing, related to state variables, such as velocity and heading direction, should be proposed.

*3.1. The Oscillatory Interference Model and Path Integration*

As previously analyzed, locations are marked by the firing of grid cells, which depends on the agent's state. The oscillatory interference model is created to relate the firing mechanism to the agent's self-motion [13]. The grid cell firing is assumed to result from two or more oscillations.

As a precondition, a baseline oscillation offers the baseline frequency $f_b(t)$ to the whole grid cell system, akin to the reference clock in conventional navigation. The others, treated as the intrinsic membrane potential oscillations (MPOs), offer the active frequency $f_a(t)$. As argued before [13–15], $f_a(t)$ varies relative to $f_b(t)$ and reflects the running speed in a preferred direction:

$$f_a(t) = f_b(t) + \beta v(t) \cos(\phi(t) - \phi_d) \tag{1}$$

where $v(t)$ and $\phi(t)$ are the speed and direction of motion, respectively. $\beta$ is a constant, regarded as the reciprocal of grid scale, and $\phi_d$ denotes the preferred direction.

It makes sense that in a fixed time interval $[0, t]$, both the running speed and direction are constant. By considering the phase difference between the baseline oscillation and the MPO, i.e., the time integration of corresponding frequency difference, we obtain:

$$\begin{aligned}
\varphi &= \int_0^t f_a(\tau) - f_b(\tau) \, \mathrm{d}\tau \\
&= \int_0^t 2\pi\beta \, v(\tau) \cos(\phi(\tau) - \phi_d) \, \mathrm{d}\tau \\
&= 2\pi\beta s(t) \cos(\phi(t) - \phi_d)
\end{aligned} \tag{2}$$

which indicates the displacement projection to the preferred direction. Hence, the above mechanism of location encoding is known as the velocity-controlled oscillator (VCO) [13]. Moreover, as a phase code, (2) exhibits cyclical fluctuations, which corresponds to the scheme of vector navigation. It is also worth noticing that the constant $\beta$ is actually determined by the spatial scale, and is named the spatial scaling factor.

Since the baseline oscillation and MPO are both manifested as sinusoidal shapes, the oscillatory interference model can be implemented by the addition of the baseline oscillation and an MPO [14].

$$P(t) = \cos(2\pi f_a(t)) + \cos(2\pi f_b(t)) \tag{3}$$

It is notable that the amplitudes of both oscillations are assumed to be unity, without loss of generality.

To be clearer, (3) can also be rewritten as:

$$P(t) = \cos(\pi \, (f_a(t) - f_b(t))) \cos(\pi(f_a(t) + f_b(t))) \tag{4}$$

Obviously, with the carrier at the mean of $f_a$ and $f_b$, the resultant oscillation outputs a signal modulated by an envelope at half the difference between $f_a$ and $f_b$. Referring to (2), the amplitude of (4) is determined by the cyclical phase code. Once the peaks are reached, the corresponding VCO fires spikes, which may signal the displacement along its preferred direction.

With $n$ VCOs, the corresponding interference models with different preferred directions, similar to (3), are combined multiplicatively to represent the firing rate of grid cells [13]; that is, the grid cell firing depends on the phase offsets caused by VCOs, which is also translated as a path integration process.

### 3.2. The Location Coding by Multiple VCOs

According to (2)–(4), a single VCO can provide location information along the preferred direction. Consequently, given a plane, a moving agent can be positioned by a pair of VCOs with different preferred directions and the same scale; that is, a 2D space can be recognized as a triangular grid for autonomous navigation.

As stated in [13–15], the grid cell firing is driven by several VCOs with different preferred directions but the same scale, coordinated by the so-called "coincident detection" [14]. For the sake of simplicity, we only consider the interference between two VCOs, whose spatial phases are provided according to (2). Although the grid cell only fires at specific locations, it is reasonable to assume that spatial phases are output all the time along the whole trajectory. This mechanism is actually the location encoding.

Thus, given the preferred direction $\phi_1(t)$ and $\phi_2(t)$, the encoding procedure is deduced from (2):

$$
\begin{aligned}
\varphi_1 &= 2\pi\left(\frac{s_1}{G} - \left\lfloor\frac{s_1}{G}\right\rfloor\right) \\
\varphi_2 &= 2\pi\left(\frac{s_2}{G} - \left\lfloor\frac{s_2}{G}\right\rfloor\right)
\end{aligned}
\tag{5}
$$

where $s_1$, $s_2$ represent displacement projections along preferred directions $\phi_1$ and $\phi_2$, respectively, $\lfloor\cdot\rfloor$ is the modulo operator, and $G$ denotes the grid scale as:

$$
G = \frac{1}{\beta}
\tag{6}
$$

Obviously, a moving agent can be localized by (5) on the triangular grid. Nevertheless, a significant issue known as the phase ambiguity still remains, since the encoding scheme is cyclical [14], which means that the proposed scheme according to (5) only works within the radius of $G$. As a result, autonomous navigation for large-scale space becomes unachievable.

## 4. Encoding Scheme for Grid-Like Self-Location

Recalling (5), let $N = \lfloor s_m(t)/G \rfloor$, where $m \in \{1, 2\}$. It is readily seen that the key point of large-scale navigation is the computation of $N$, which is known as integer ambiguity. Here, motivated by the idea of the "Three Carrier Ambiguity Resolution (TCAR)" algorithm [19], a phase unwrapping algorithm is proposed.

### 4.1. Vector Navigation Based on Step-Wise Phase Unwrapping

Thanks to the special architecture of mammals' mEC, a grid cell network is supposed to provide the capacity of multi-scale periodic representation of self-location, resulting in the solution of phase ambiguity [12].

In this section, we propose a novel algorithmic level description to model the biological mechanism of ambiguity resolution.

Take the 1D case into consideration. By employing the oscillatory interference model, the current location is encoded by two VCOs with scales $G_i$ and $G_j$, respectively:

$$
\begin{aligned}
\varphi_i &= 2\pi(s/G_i - N_i + \epsilon_i) \\
\varphi_j &= 2\pi\left(s/G_j - N_j + \epsilon_j\right)
\end{aligned}
\tag{7}
$$

where $N_i$, $N_j$ represent integer ambiguities. In contrast with (5), the phase noises $\epsilon_i$, $\epsilon_j$, resulting from the random perturbation of velocity inputs are taken into account.

According to (7), by investigating the phase difference between $\varphi_i$ and $\varphi_j$, we obtain:

$$
\varphi_d = \varphi_i - \varphi_j = 2\pi(s/G_d - N_d + \epsilon_d)
\tag{8}
$$

where:

$$
\begin{aligned}
G_d &= \frac{G_i G_j}{G_i - G_j} \\
N_d &= N_i - N_j
\end{aligned}
\tag{9}
$$

which seems like an equivalent 1D path integration with scale $G_d$. From (9), it implies that $G_d$ is much larger than either $G_i$ or $G_j$, when $G_i$ and $G_j$ have close values, which means the navigation can reach a much larger range. Moreover, in this case, the equivalent integer ambiguity $N_d$ is close to 0. This means that, if $G_i$ and $G_j$ are appropriately selected, the large-scale navigation without any phase ambiguity is achievable.

Thus, the multi-scale periodic code $[\varphi_i, \varphi_j]$ can be easily decoded as:

$$\hat{s} = G_d \varphi_d / (2\pi) \tag{10}$$

However, due to the phase noise $\epsilon_d$, the estimate error of (10) is non-negligible. Moreover, by investigating (8) and (10), it is obviously seen that when $G_d$ becomes larger, the phase noise may have greater effect on the accuracy of estimation.

Inspired by the idea of TCAR, a step-wise phase unwrapping (SPU) algorithm is proposed to satisfy large-scale navigation while ensuring acceptable estimate accuracy. In this method, $n$ VCOs are employed, corresponding to $m = C_n^2$ equivalent scales, where $C_n^2$ denotes the number of possible combinations among $n$ scales. Next, choose the largest equivalent scale $G_{d_1}$ and calculate the displacement estimate $\hat{s}_1$ according to (10). Then, switch to the second-largest equivalent scale $G_{d_2}$, and calculate the integer ambiguity $N_{d2}$, by substituting $\hat{s}_1$ into (8). Thus, a more accurate $\hat{s}_2$ can be obtained. Estimates depending on other equivalent scales are derived similarly, in a recursive way, and the influence of phase noise can be relieved in a step-wise way. More details are shown in Algorithm 1.

---

**Algorithm 1** Step-Wise Phase Unwrapping for 1D Path Integration

---

**1: Input:** Selected scale set $\{G_1, G_2, \cdots, G_n\}$, where $G_i > G_{i+1}$
       Corresponding phase vector $\{\varphi_1, \varphi_2, \cdots, \varphi_n\}$
**2: Output:** $\hat{s}$
**3:** Calculate equivalent scales by (9) and sort them in descending order
    $\{G_{d_1}, G_{d_2}, \cdots, G_{d_m}\}$, where $G_{d_i} > G_{d_{i+1}}$
    List the responding set of phase difference set $\{\varphi_{d_1}, \varphi_{d_2}, \cdots, \varphi_{d_m}\}$
**4:**   $\hat{s}_1 = G_{d_1} \varphi_{d_1} / (2\pi)$
**5: While** $i < m - 1$ **Do**
$$N_{d_{i+1}} = [\hat{s}_i / G_{d_{i+1}} - \varphi_{d_{i+1}} / (2\pi)]$$
    $\hat{s}_{i+1} = G_{d_{i+1}} (\varphi_{d_{i+1}} / (2\pi) + N_{d_{i+1}})$
**6:** $N_{d_{m+1}} = [\hat{s}_m / G_n - \varphi_n / (2\pi)]$
**7:**   $\hat{s} = G_n (\varphi_n / (2\pi) + N_{dm+1})$

---

In a 2D plane, Algorithm 1 is simultaneously performed in two different directions, and then the localization is achieved by the vector sum. In summary, the self-location is performed by $n$ grid cells with different spatial scales, but only two preferred directions $\phi_1$ and $\phi_2$ are engaged. Any considered position can be encoded by several equivalent scales, as $[\varphi_{d_{1,1}}, \varphi_{d_{1,2}}, \cdots, \varphi_{d_{1,n}}]$ along direction $\phi_1$, and $[\varphi_{d_{2,1}}, \varphi_{d_{2,2}}, \cdots, \varphi_{d_{2,n}}]$ along direction $\phi_2$, and decoded through the SPU algorithm.

*4.2. The Proposed Scheme for Autonomous Navigation*

It is believed that self-location of mammals depends on the firing of grid cells, which results from the interference of VCOs. However, no matter how the grid cell actually works in the sense of biology, in our proposed AI, based on the AC architecture, the grid-like module is considered to be composed of several direction-sensitive elements with different spatial scales, performing functions of VCOs. For the $i$ th element, the current velocity and running direction constitute the input vector, while the corresponding output is the location code $\{\varphi_{1,i}, \varphi_{2,i}\}$ with scale $G_i$. As previously mentioned in Section 2, phase vectors $\phi_1$ and $\phi_2$ are formed by all of the elements and provided to a certain unit, such as an FPGA, on which the large-scale self-location is implemented through the SPU algorithm. The flowchart of the proposed scheme is shown in Figure 2.

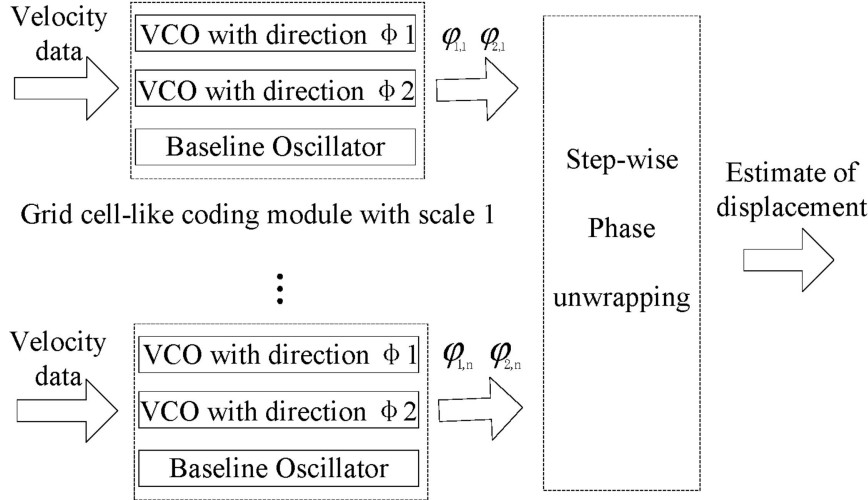

**Figure 2.** Flowchart of the proposed autonomous navigation.

### 5. Numerical Results

In this section, simulation results are presented to evaluate the performance of the proposed autonomous navigation algorithm. The scenarios of four grid-like modules, corresponding to four respective spatial scales, are considered. For each module, the preferred directions are set as 0 and $\pi/3$, i.e., a plane coordinate is constituted by two axes with angle $\pi/3$. Since the SPU strategy is adopted, the selection of combinations of different spacial scales may significantly influence the performance of self-location. As demonstrated in [16], spatial scales of grid cells usually range from 25 cm to 300 cm. Hence, three groups of scales, set as case 1: {50 cm, 50.1 cm, 51 cm, 60 cm}, case 2: {60 cm, 60.1 cm, 61 cm, 70 cm}, and case 3: {101 cm, 101.1 cm, 102 cm, 60 cm}, are considered. All the simulations were carried out in MATLAB 2018 software.

Supposing that in each step the perturbation of the phase code is 2% of the corresponding integer ambiguity, according to Equations (7), (8) and (10) and the definition of standard derivation, the error analysis is summarized in Table 1. It is readily seen that with any scale combination, the standard deviation decreases step-wise as the equivalent scale decreases.

**Table 1.** Error analysis.

|  | Step 1 | | Step 2 | | Step 3 | | Step 4 | | Step 5 | | Step 6 | |
|---|---|---|---|---|---|---|---|---|---|---|---|---|
|  | $G_{d1}$ | $\delta_1$ | $G_{d2}$ | $\delta_2$ | $G_{d3}$ | $\delta_3$ | $G_{d4}$ | $\delta_4$ | $G_{d5}$ | $\delta_5$ | $G_{d6}$ | $\delta_6$ |
| Case 1 | 250.5 | 7.09 | 28.4 | 0.80 | 25.5 | 0.72 | 3.4 | 0.10 | 3.0 | 0.09 | 0.5 | 0.01 |
| Case 2 | 360.6 | 10.1 | 40.7 | 1.15 | 36.6 | 1.04 | 4.7 | 0.13 | 4.2 | 0.12 | 0.6 | 0.02 |
| Case 3 | 1001.0 | 28.31 | 112.3 | 3.18 | 101.0 | 2.86 | 12.3 | 0.35 | 11.0 | 0.31 | 1.0 | 0.03 |

The AA is assumed to move at a constant speed. The moving direction randomly switches per second with the values ranging from 0 to $\pi/2$. Hence the self-location is performed per second. The phase noise is assumed to be normally distributed, with mean $\mu$, the actual phase, and standard deviation $\delta$, 2% of $\mu$. The estimation performance is evaluated by the standard derivation (SD). Results are obtained through 1000 Monte Carlo simulations in an observation window of 100 s. First, the agent is supposed to move at a speed of 0.5 m/s. As shown in Figure 3, it ultimately achieves a displacement of 45.1 m. The scale combination in case 3 performs significantly better than in the other two.

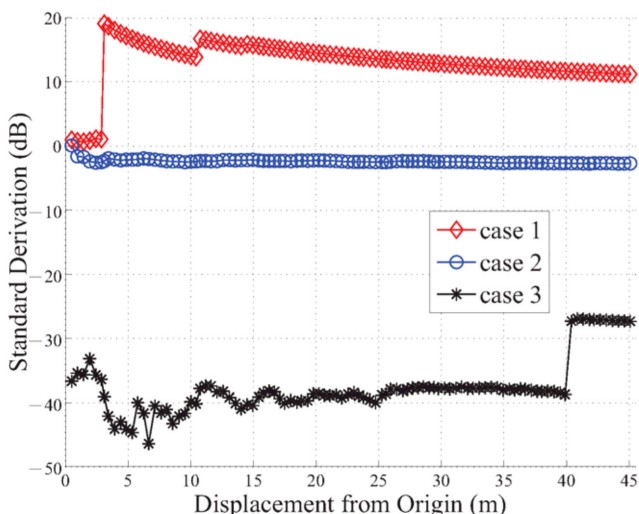

**Figure 3.** Performance at a speed of 0.5 m/s.

Subsequently, the moving speed decreases to 0.3 m/s, within the observation window of 500 s. As demonstrated in Figure 4, performances in case 1 and case 2 both achieve visible improvements. In particular, in the range of 25.7 m, the combination in case 2 performs as well as in case 3. As the AA moves further, the performance of case 2 becomes worse. Meanwhile, the performance in case 3 is always more robust and better than in the others.

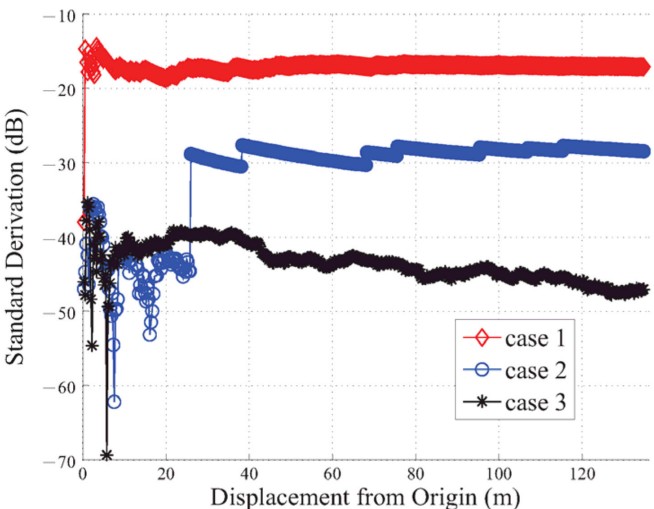

**Figure 4.** Performance at a speed of 0.3 m/s.

These consequences show that the performance of the proposed algorithm deeply depends on the selection of spatial scale combination. Although the estimate precision seems better in the case of a small scale, a larger phase output causes larger integer ambiguity, which deeply affects the calculation of $N_d$.

As validation, the trajectory in case 3 is also sketched in Figure 5, and 10 positions are marked. It is obvious that 1000 estimates scatter closely around each actual position, which implies the proposed algorithm works reasonably well.

As can be readily seen, the proposed SPU extends the autonomous navigation to a significantly long range, and its performance only depends on the phase noise and scale combination. However, another method exists to address the issue of long-range navigation: in [17], the modular arithmetic (MA) scheme is proposed and the spatial range is extended to the product of the involved scales. It should be noted that in this scheme, spatial scales must be coprime integers, which is not necessary for the SPU scheme. Correspondingly,

under the idea of the MA scheme, Chinese remainder theorem is adopted to solve phase ambiguity [20]. With the observation window of 100 s, the performance of these schemes is compared at the moving speed of 0.3 m/s, in the absence of noise. As shown in Figure 6, the SPU scheme performs better and is more robust. Additionally, as previously mentioned, the SPU scheme is more flexible since the spatial range of the MA scheme is restricted within the product of involved scales.

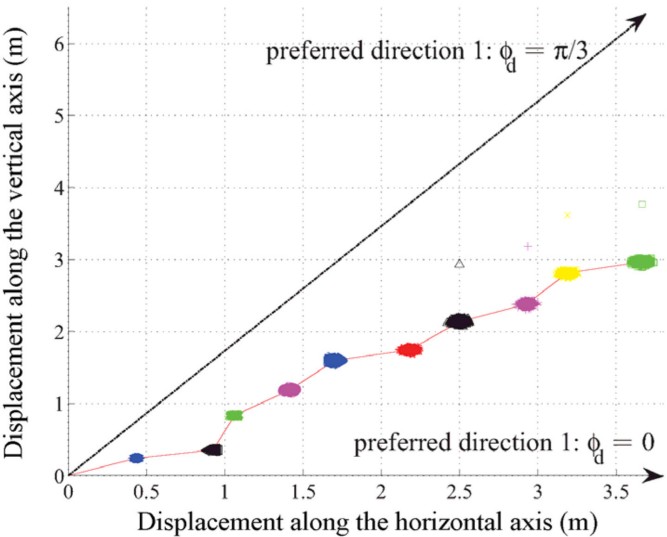

**Figure 5.** Trajectory of an AA.

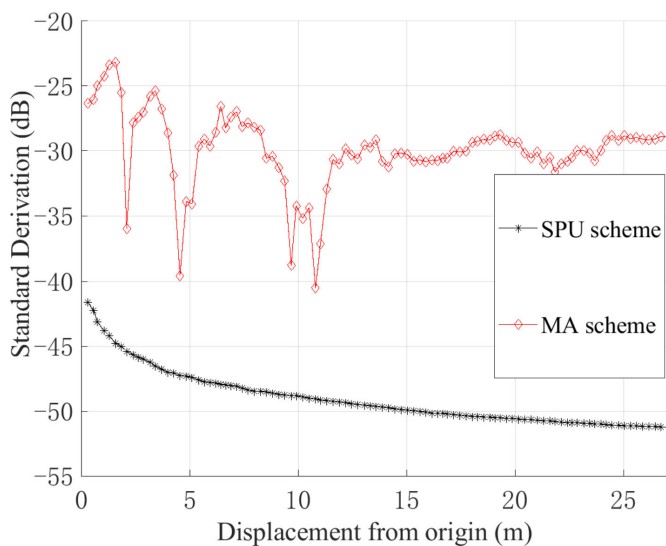

**Figure 6.** Comparison between the SPU and MA schemes.

## 6. Conclusions

In this paper, we concentrate on the autonomous navigation problem of AAs in a GNSS/radio-denied environment. By borrowing the idea of the "cognitive map" that lies in a mammal's brain, an autonomous localization system is built based on grid-like modules with different spatial scales. These modules are direction-sensitive and controlled by moving velocities. The temporal location of a moving agent is represented by phase codes. Using the proposed SPU algorithm, the large-scale self-location is implemented without any phase ambiguity. Simulations also validate the performance of the proposed self-location scheme. However, the accuracy of self-location is still significantly affected by the selection of combined grid module scales. As a result, we will concentrate on the optimization of module scale selection in future work, as well as the validation in real environments.

**Author Contributions:** Conceptualization, C.D.; methodology, L.X.; software, L.X.; validation, C.D. and L.X.; formal analysis, C.D.; investigation, L.X.; resources, C.D.; writing—original draft preparation, L.X.; writing—review and editing, L.X.; visualization, L.X.; supervision, C.D.; project administration, C.D.; funding acquisition, C.D. All authors have read and agreed to the published version of the manuscript.

**Funding:** This research was funded by the National Natural Science Foundation of China, grant number 61973314.

**Acknowledgments:** The authors would like to thank the anonymous reviewers for their careful review and constructive comments.

**Conflicts of Interest:** According to the policy as well as our moral obligation, we declare that there are not any relevant conflicts of interest.

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
