# Peer review of "Self-Location Based on Grid-like Representations for Artificial Agents"

_electronics, doi:10.3390/electronics12122735_

Round 1
Reviewer 1 Report
This study focuses on developing a grid-like navigation model for AI applications. The firing patterns of grid cells are analyzed to propose an oscillatory interference model for encoding location information. Additionally, a multi-scale periodic representation mechanism is constructed to enable large-scale navigation. To mitigate phase ambiguity resulting from grid cell firing patterns, a step-wise phase unwrapping algorithm is introduced.
The following comments are for the authors to improve the paper and to clarify some concepts.
1. 1. In your abstract, you mentioned that modeling dynamic and complex scenes for effective environment representation remains an issue. Could you elaborate in the introduction on some of the challenges faced in accurately modeling such environments for artificial agents?
2. In the introduction, a literature review is presented. However, the comments on existing results could be more critical so as to motivate the problem under study. For example, what's the main problem compared with the existing results? More discussions can be added.
3. Current solutions for self-location often rely on GNSS, INS, or vision/laser-aided localization. However, you mentioned that these approaches have limitations in terms of computational burden and effective environment mapping. Could you explain these limitations in more detail and highlight the specific challenges faced in achieving effective environment mapping?
4. You mentioned that current neural network implementations for large-scale navigation often result in high computational consumption. With the high computational power available these days, could you provide more insights into these potential neural network implementations and explain why they may not be feasible for conventional artificial agents?
5. In Introduction section, please add a summary of the main contributions and implications of your work on autonomous navigation in a GNSS/radio-denied environment? How does the proposed self-location scheme improve the performance of artificial agents in such environments?
6. Throughout the paper, there are many grammar error and typos, which should be removed.
7. The authors should add on the advantages of the proposed method over others work.
8. In Figure 1, the outputs of the grid-like module and vision module are provided as network weights for the A3C learner. Explain how these network weights are used in the A3C architecture? How does the extrinsic reward of the previous action from the environment influence the learning process?
9. Could you provide some examples or insights into the practical application of the proposed framework in real-world scenarios? How does the integration of the grid-like module and vision module contribute to the overall performance of artificial agents in autonomous navigation tasks?
10. Inspired by the computational function of grid cells, you propose transplanting this function into the AI framework. How can a grid-like module be implemented using electronic circuits? How does this implementation enable autonomous navigation and integrate with the Markov inference process?
11. The problem presented in Figure 1is not clear. More explanation is needed.
12. Adjust the equation numbering location, such as equations (3), (4), 7, 9, 10,
13. In line 227, Cn is not defined.
14. In line 244, AI is mentioned, but it is not explained in the paper what type of AI tool is used and where.
15. The error analysis in Table 1, is not explained how it is found.
16. The x-y axis in Figure 4 is not labeled nor units are specified.
17. The comparisons with existing results should be provided.
Reviewer 2 Report
Dear Authors,
Please try to enhance your abstract, in the way of answering several questions for the title of the paper as; what is Self-location Based on Grid-like Representations for Artificial Agents,then why, the. How,....and the expected results, then hint for the importance of these results.
In addition, you must improve the introduction, it seems not sufficient.
Besides, please review all the figures and tables, they must be improved by high level comments and additions, also the numbering of the figures "figure 4 was twice " please take care and improve.
All the paper must be checked for grammar errors.
Reviewer 3 Report
The paper focuses on the self-location of the robot. It proposes a grid-like navigation model for artificial intelligence applications based on the firing patterns of grid cells for autonomous navigation. However, there are several areas that require further explanation and improvement.
1) The paper lacks a comparison with previous methods. It is important to provide a comparative analysis to demonstrate the advantages and novelty of the proposed approach.
2) There is no explanation regarding the setup of the simulations. It is necessary to provide details about the simulator used and the specific configuration employed.
3) The paper lacks tests in real-world scenarios. Conducting experiments in real environments would strengthen the credibility and practicality of the proposed algorithm.
4) The paper does not clearly explain the sensors in the research. Additionally, the proposed algorithm fails to specify any particular sensors that can be employed in the implementation.
5) In Figure 1, three modules are depicted, but the sensors associated with each module are not specified. Additionally, the paper does not clarify the meaning of extrinsic and intrinsic rewards, nor does it explain the consequences of receiving positive rewards or being penalized.
6) In Equation 1, there is no explanation of the variable beta. The authors simply state that it is a constant, but they fail to provide any information regarding its representation and relevance within the model.
7) The paper does not discuss any limitations of the proposed algorithm. It is essential to address the potential drawbacks or constraints of the method to provide a comprehensive understanding of its applicability.
Reviewer 4 Report
The author proposed a self-location approach based on grid-like representations for artificial agents. Overall, this paper is well-organized, but the innovation and comparison with similar algorithms need to be enhanced.
1. Abstract: The research background is too much, which needs to be simplified. The contributions of this work need to be rewritten in a more technical way. Then the description of detailed experimental results is required.
2. Introduction part: the existing AI models are recommended to add to this part. In addition, a comparison with similar algorithms is required. The last paragraph of this part can be divided into two paragraphs, one indicates the contributions of your work, then is the paper structure.
3. It is advised to add an algorithm diagram to describe the whole structure.
4. The experimental part needs to be enhanced, the comparison with existing state-of-art algorithms is missing.
5. Conclusion part: The advantages and disadvantages proposed approach are recommended to add, and the future work.
n/a
Round 2
Reviewer 3 Report
Overall, the authors should provide more detailed and specific responses to the comments, addressing the concerns raised by the reviewers. All of the author's responses have not been clear and are not properly detailed in the 'author's response' document. For example:
Comment 2: The author's response states, 'Point taken. The relevant information has been added in the revised manuscript.'
Comment 4: The author mentions that the related content has been supplemented at the end of Section 2.
Comment 6: The author states that an explanation has been added in the following section.
Additionally, the paper lacks experimental evidence. As a result, there is insufficient support to suggest that the research conducted is suitable for publication in a journal.
Reviewer 4 Report
The author has solved all my concerns in the revised version, hence, I am glad to recommend this paper to be accepted.
n/a
Author Response
We are very glad to hear that and thanks for your approval.
Round 3
Reviewer 3 Report
No comments